# Gene Expression Signature of Acquired Chemoresistance in Neuroblastoma Cells

**DOI:** 10.3390/ijms21186811

**Published:** 2020-09-16

**Authors:** Mohamed Jemaà, Wondossen Sime, Yasmin Abassi, Vito Alessandro Lasorsa, Julie Bonne Køhler, Martin Michaelis, Jindrich Cinatl, Mario Capasso, Ramin Massoumi

**Affiliations:** 1Department of Laboratory Medicine, Translational Cancer Research, Lund University, Medicon Village, 22381 Lund, Sweden; mohamed.jemaa@med.lu.se (M.J.); wondossen.sime@med.lu.se (W.S.); yasmin.abassi@med.lu.se (Y.A.); julie.bonne_kohler@med.lu.se (J.B.K.); 2Dipartimento di Medicina Molecolare e Biotecnologie Mediche, Università degli Studi di Napoli Federico II, Via Sergio Pansini 5, 80131 Naples, Italy; lasorsa.alessandro@gmail.com; 3CEINGE Biotecnologie Avanzate, Via G Salvatore, 80131 Naples, Italy; 4School of Biosciences, University of Kent, Canterbury CT2 7NJ, UK; M.Michaelis@kent.ac.uk; 5Institute of Medical Virology, Clinics of the Goethe-University, D-60596 Frankfurt am Main, Germany; cinatl@em.uni-frankfurt.de; 6IRCCS SDN, Via Emanuele Gianturco, 113, 80143 Naples, Italy

**Keywords:** chemoresistance, neuroblastoma, vincristine

## Abstract

Drug resistance of childhood cancer neuroblastoma is a serious clinical problem. Patients with relapsed disease have a poor prognosis despite intense treatment. In the present study, we aimed to identify chemoresistance gene expression signatures in vincristine resistant neuroblastoma cells. We found that vincristine-resistant neuroblastoma cells formed larger clones and survived under reduced serum conditions as compared with non-resistant parental cells. To identify the possible mechanisms underlying vincristine resistance in neuroblastoma cells, we investigated the expression profiles of genes known to be involved in cancer drug resistance. This specific gene expression patterns could predict the behavior of a tumor in response to chemotherapy and for predicting the prognosis of high-risk neuroblastoma patients. Our signature could help chemoresistant neuroblastoma patients in avoiding useless and harmful chemotherapy cycles.

## 1. Introduction

Neuroblastoma (NB) is a childhood solid tumor originating from undifferentiated neural progenitor cells of the sympathetic nervous system. The disease accounts for 8–10% of all childhood cancers and 15% of pediatric cancer deaths [1,2,3]. Different criteria, such as patient age, *MYCN* amplification, chromosomal aberrations, and history of metastasis, are used in deciding patient treatment [4]. Available treatment strategies for NB patients include chemotherapy followed by surgical resection, myeloablation and autologous stem cell rescue, radiation, and immunotherapy [5,6]. Common chemotherapeutics for NB includes cyclophosphamide, cisplatin, doxorubicin, etoposide, carboplatin, and vincristine, either in combination or alone [7,8,9]. Compared to low- and intermediate-risk NB, it is often challenging to successfully treat high-risk patients. Particularly, recurrence is observed in more than 50% of children with high-risk disease [4]. Previous studies have identified that NBs undergo mutational evolution during therapy, and that relapsed disease can be driven by targetable oncogenic pathways including the RAS-MAPK pathway or epithelial-mesenchymal transition [10,11,12]. Recently it was discovered that NBs consist of two types of tumor cells; undifferentiated mesenchymal and committed adrenergic cells, where mesenchymal cells showed characteristics of chemoresistant cells [13]. Since options for tackling chemoresistant NB in clinical practice are limited, there is an urgent need for developing additional treatment strategies.

Vincristine is a common chemotherapeutic drug used against NB and vincristine is included in the rapid COJEC (Cisplatin [C], vincristine [O], carboplatin [J], etopoide [E], and cyclophosphamide [C]) treatment regimen. Vincristine is a vinca alkaloid that blocks cell growth by interfering with mitosis. By binding to β-tubulin, vincristine disrupts normal microtubule dynamics and destroys mitotic spindles [14,15,16]. This leads to mitotic cell cycle arrest followed by cell death. In addition to the general mechanisms that can cause drug resistance, altered expression of tubulin isotypes can also contribute to the resistance of cells to tubulin-binding drugs [17]. In the present study, we identify chemoresistance gene expression signatures that could predict the behavior of a tumor in response to chemotherapy independently of current prognostic factors, including age at diagnosis, stage, and *MYCN* status. Our results strongly support the role of the selected genes in chemoresistant mechanisms and in poorer prognosis in NB.

## 2. Results

In the present study, we established several vincristine-resistant NB cell lines to investigate the influence of vincristine resistance on cell growth and biological behavior, as well as to identify a novel inhibitor that can limit the growth of these cells. We decided to use the NB cell line SK-N-BE(2)-C, which is derived from a metastatic site (bone marrow) following treatment relapse, and harbors MYCN amplification, p53 mutation (C135F), 1p deletion, and without ALK mutations. Be2c cells that were sensitive to vincristine treatment (1 ng/mL), were cultured for 7 months with gradually increasing concentrations of vincristine (1–10 ng/mL, Figure 1A) followed by the generation of cell lines resistant to even higher concentrations of vincristine (15–50 ng/mL, Figure 1A). The established vincristine-resistant cells (Be2c-VCR) remained resistant even when cultured in the absence of vincristine for 3 weeks and subsequent treatment with vincristine, suggesting that the mechanism of drug resistance is not immediately reversible upon cell release from drug (Figure 2A,B). Changes in cytoskeleton structure have previously been linked to vincristine resistance [18,19]. To assess the cytoskeleton of our resistant cells, we examined cell morphology by α-tubulin immunofluorescent staining. The resistant and parental cells were similar in size (Figure 1B and Figure 2C), whereas Be2c-VCR cells were more elongated in contrast to parental cells that showed a more rounded phenotype (Figure 1C). Elongated cells or neurite outgrowth in general can be considered to be neuroblastoma cell differentiation and a non-proliferative stage of the cells. In this aspect, we compared the degree of cell survival and cell proliferation between Be2c-parental and Be2c-VCR cells. PI staining revealed no difference in the number of cells undergoing cell death between resistant and parental cells (Figure 1D). Analyzing cell growth, the resistant cells showed no differences in proliferation as compared with parental cells (Figure 1E and Figure 2D). Reduced proliferation is usually seen for cancer cells upon prolonged administration of anticancer therapy or when cells are becoming less sensitive to chemotherapy [20]. Cell cycle analysis demonstrated that no major differences could be observed in the cell cycle phases between parental and resistant cells (Figure 1F and Table 1). In contrast, the colony formation assay showed that resistant cells generated an increased number of clones as compared with parental cells (Figure 1G). In addition, colonies formed by Be2c-VCR (20 ng/mL) cells were significantly larger as compared with parental cells (Figure 1H).

Next, we investigated whether the vincristine-resistant cells were able to grow in reduced serum conditions. While both resistant and parental cells were unable to proliferate in the absence of serum (0% serum, Figure 3A), resistant cells could grow and divide in reduced serum media (2.5%), which was not observed for parental cells (Figure 3A). Subsequently, we explored whether the vincristine-resistant cell lines were also resistant to other microtubule-destabilizing drugs such as nocodazole and paclitaxel. Time-dependent studies of nocodazole and paclitaxel treatments demonstrated that Be2c-VCR cells were more resistant to these drugs than parental cells (Figure 3B,D). Assaying cell death by 7-AAD staining also revealed a higher percentage of dead (7-AAD positive) cells for paclitaxel-treated parental cells (Figure 3C, left panel) than for the vincristine-resistant cells (Figure 3C, middle and right panels). In addition, the half maximal inhibitory concentration (IC50) values of other chemo agents clinically used for NB treatment, such as etoposide and cisplatin, were also slightly higher for VCR-resistant cells (Table 2).

To identify other possible mechanisms underlying vincristine resistance in Be2c cells (VCR-20), we investigated the expression profiles of 84 genes known to be involved in cancer drug resistance. The expression levels of 26 genes involved in the mechanism of vincristine resistance were identified (Figure 4A), which included MVP (major vault protein), TOP2-α (topoisomerase [DNA] II-α), TOP2-β, and DHFR (dihydrofolate reductase) among the top genes to be highly upregulated in vincristine resistant cells as compared with parental cells. We next sought to assess if a signature (hereafter chemoresistance signature) based on the vincristine resistance genes found in Be2c cells could predict the patient outcome. The following two well-annotated public data sets of gene expression on NB tumors were considered in the analysis: GSE16476 (88 samples, Affymetrix U133P2 microarrays) and GSE62564 (498 samples, RNA-Seq). The list of 26 genes from the chemoresistant signature was used to perform k-means clustering analysis on the cohorts as a whole. Given that the information on patient’s therapy history was missing and since non-high-risk patients could also undergo chemotherapy treatment (1), we decided to keep all the individuals of each cohort. For both datasets, we found that expression values of the queried genes could group the samples in two clusters with distinct characteristics. The heat map for the GSE16476 data, after the clustering steps, is depicted in Figure 4B. We named chemoresistant POS groups (*n* = 42 and *n* = 168 for GSE16476 and GSE62564, respectively), which were the clusters with an overall expression pattern coherent with that of the chemoresistance signature, whereas chemoresistant NEG (*n* = 46 and *n* = 330 for GSE16476 and GSE62564, respectively) were the groups with an opposite gene expression pattern. The chemoresistant POS group is composed of samples with worse clinical features as compared with chemoresistant NEG (Figure 4C and Figure 5A). For GSE16476, we found MYCN amplification in 33% of chemoresistant POS and in 2% of chemoresistant NEG. In GSE62564, N-MYC amplification was in 52% of chemoresistant POS and in 2% of chemoresistant NEG. Risk status classification for GSE62564 was provided as follows: 76% were high-risk in chemoresistant POS and 2% in chemoresistant NEG. On the basis of phenotypic characteristics of the two classes in each dataset, we produced overall and relapse-free Kaplan–Meier survival curves. Figure 4D and Figure 5B show that chemoresistant POS groups have worse overall survival probabilities as compared with chemoresistant NEG, as well as worse relapse-free survival probabilities over time.

In addition, chemoresistant gene signature predicts outcome independently of current prognostic factors, including age at diagnosis, International Neuroblastoma Staging System (INSS) stage, and MYCN status in both datasets (Table 3). We found specific gene expression patterns that could predict the behavior of a tumor in response to chemotherapy. Indeed, our signature could help chemoresistant POS patients (76% of high-risk patients and 24% of non-high-risk patients, Figure 4C) in avoiding useless and harmful chemotherapy cycles. Our results strongly support the role of the selected genes in vincristine resistant mechanisms and in poorer prognosis in the NB cohorts we surveyed.

## 3. Discussion

Treatment of metastatic NB disease continues to be challenging, in particular in children with high-risk disease who have low survival despite aggressive multimodal treatment strategies (1). Despite surgery, high dosage chemotherapy combined with myeloablative cytotoxic therapy, autologous haematopoietic stem cell transplantation, and radiotherapy, the prognosis of these patients still remains poor. One of the major factors that contributes to chemotherapeutic treatment failure of high-risk NB patients is the acquisition of drug resistance by tumor cells, and this failure highlights the importance of developing novel approaches in this field. Since MYCN amplification is strongly associated with rapid progression and poor prognosis in NB [21,22], we decided to generate drug resistance in MYCN amplified cell line derived from NB patient with a history of aggressive and metastasis (SK-N-BE(2)-C. Beyond p53 mutations, Be2C cells harbor other important pathogenic genomic aberrations such as 1p21.1-36.3 deletion, which is a well-known positive marker of unfavorable outcome in NB, as well as 9p21-24.3 and 3p22.1-25.3 loss, which is found in other tumors [23,24]. Moreover, large scale homozygous loss of chromosome 3p is a common event in NB, and it is associated with tumors from older children [25,26].

Previous studies have shown that drug resistance-induced phenotypes led to an increase in metastatic potential and changes in the biological characteristics of NB cells [27,28]. Vincristine-resistant cells generated in our study showed changes in their morphology and cytoskeleton as compared with the parental cells. A previous report using vincristine-resistant sublines of the NB cell lines UKF-NB-2 and UKF-NB-3, showed that resistant cells harbor penetration capacity [29] and were more proficient in clonal growth [30]. This was in accordance with our colony assay, where we found vincristine-resistant cells formed more and bigger colonies. Furthermore, vincristine-resistant cells, but not parental cells, could survive and even proliferate under conditions of reduced serum concentration. One possible explanation for this could be the ability of the cells to adapt and change their cellular metabolism under serum deprivation or even upon treatment with anticancer drugs. The vincristine-resistant NB cells also exhibited cross-resistance to a number of other drugs used in NB treatment. Such multidrug resistance has also been observed in highly malignant NBs derived from patients at different phases of therapy [31].

Investigating the mechanism of vincristine resistant of our Be2c-VCR NB cells, it was found that genes encoding for epidermal growth factor receptor (EGFR), dihydrofolate reductase (DHFR), microsomal epoxide hydrolase (EPHX1), MVP, cytochrome P450 3A4, and DNA topoisomerase II alpha and beta (TOP2A and TOP2B) were highly upregulated in Be2c-VCR cells as compared with the parental Be2c cells. The importance of these differentially upregulated genes has been well established in cancer drug resistance mechanisms and tumor progression [32,33,34,35,36]. In particular, high expression of EGFR could be associated with RAS-MAPK pathway activation identified as a signaling pathway in relapsed neuroblastomas [10,12]. However, further studies need to establish the precise mechanism behind chemoresistant neuroblastoma cells.

In this work, we investigated the expression profiles of 84 genes known to be involved in cancer drug resistance in VCR-resistant NB cells. Starting from 26 differentially expressed genes, we built a gene signature and found that large portions of patients in the cohorts we surveyed (33.7% and 47.7% for GSE62564 and GSE16476, respectively) could not benefit from vincristine treatment. Indeed, these subgroups of patients showed low rates of Overall Survival (OS) and Relapse-Free Survival probabilities(RFS). This is relevant in the clinical management of NB patients because the quantification of the expression levels, in NB tissue, of these 26 genes before starting the chemotherapeutic treatment could predict the drug response and avoid severe sides effects for the patients [37], and thus could help oncologists choose the most safe and effective treatment.

To our knowledge, no study has investigated the predictive potential of the expression levels of genes obtained from NB cell lines resistant to a specific chemotherapeutic agent. In the past, we and others have proposed gene-based classifiers but specifically generated to stratify high-risk or ultra-high-risk tumors [38,39,40,41]. Another study found a pivotal role of cytoskeleton and transport in vincristine resistance by in-silico reconstruction of protein networks and by using the concept of synthetic lethality and protein hubs [42].

In summary, the identified gene expression patterns could predict the behavior of a tumor in response to chemotherapy and for predicting the prognosis of high-risk NB patients. Our signature can further help chemoresistant NB patients in avoiding harmful chemotherapy cycles.

## 4. Materials and Methods

### 4.1. Cell Culture

The human NB cell line SK-N-BE(2)-C (ATCC, CRL-2268) was cultured at 37 °C and 5% CO_2_ in minimum essential medium (MEM, HyClone, Thermo Fisher Scientific, MA, USA) supplemented with 10% fetal bovine serum (FBS) (Sigma-Aldrich Sweden AB, Stockholm, Sweden) and 1% penicillin/streptomycin (GIBCO, MA, USA). To generate vincristine-resistant cells (Be2c-VCR), SK-N-BE(2)-C cells were cultured for 7 months with gradually increased concentrations of vincristine ranging from 1 to 10 ng/mL. The 10 ng/mL Be2c-VCR cells were each treated further with escalating concentrations (15–50 ng/mL) of vincristine monthly.

### 4.2. Immunofluorescence and Confocal Microscopy

Cells were cultured on coverslips in 6-well plates for 24 h, and then fixed in 100% methanol for 10 min, at −20 °C. Next, the cells were incubated in 10% FBS-PBS for 1 h at room temperature with anti-α-tubulin-FITC antibody (Clone DM1A, #F2168-2ML, Sigma-Aldrich Sweden AB, Stockholm, Sweden) and DAPI for DNA staining. Then, coverslips were washed and mounted in fluorescence mounting medium (Dako, Santa Clara, CA, USA). The images were obtained using a Zeiss LSM710 confocal microscope.

### 4.3. Proliferation and Viability Assay

Cells were seeded in 96-well plates with a 5000 cells/well density. Twenty-four hours later, cells were treated with different concentrations of nocodazole, paclitaxel, etoposide, doxorubicin, or cisplatin. After 48 h, 10 μL of alamarBlue reagent (ThermoFisher) was added to each well, mixed, and then incubated for 4 h, at 37 °C, in a CO_2_ incubator. The absorbance of each sample was measured using a scanning microplate spectrophotometer reader (Synergy 2, Biotek, Germany) by absorbance detection at 570 nm or fluorescence detection at excitation and emission wavelengths of 540–570 and 580–610 nm, respectively. The IC50 values for each treatment were calculated using the GraphPad software.

### 4.4. Clonogenic Assays

To evaluate clonogenic survival, cells were seeded at different concentrations (from 100 to 2000 cells/well) in 6-well plates. Cells were kept in standard culture conditions. Then, colonies were fixed with 4% PFA, stained with an aqueous solution of 1% (*w*/*v*) crystal violet, and counted. Only colonies made up of >30 cells were included in the quantification. To evaluate clonogenic potential, cells were seeded at low concentrations (100 cells/well) in 6-well plates and cultured in standard conditions for 15 days. Then, colonies were fixed with 4% PFA, stained with an aqueous solution of 1% (*w*/*v*) crystal violet, and counted. Colonies made of >30 cells were included in the quantification, and colony size was evaluated using the ImageJ software.

### 4.5. Apoptosis and DNA Fragmentation Assay

For apoptosis analyses, cells were fixed in PFA on coverslips and stained with a Vindelöv solution containing propidium iodide (PI). After washing, the coverslips were mounted onto glass slides and cells were examined by fluorescence microscopy. Cells were scored for apoptosis based on nuclear morphology. Apoptosis was further evaluated using a NucleoCounter NC-3000 (Chemometec, Allerod, Denmark) in conformity with the DNA fragmentation assay. Cells grown in 6-well plates were harvested by trypsinization and pooled with the cells floating in the medium. After a short centrifugation, the supernatant was removed, and precipitated cells were washed once with PBS. After a second centrifugation, cells were resuspended in a small volume of PBS and the single-cell suspensions were added to 70% ethanol for fixation. The samples were vortexed and stored for 12–24 h, at −20 °C. The ethanol-suspended cells were centrifuged and the ethanol carefully decanted. Cells were washed once with PBS, and then resuspended in NucleoCounter Solution 3 (1 µg/mL DAPI, 0.1% Triton X-100 in PBS) followed by incubation for 5 min, at 37 °C. Ten microliters of samples were loaded into a slide chamber (NC-slide A8) and the DNA fragmentation protocol was employed according to the manufacturer’s instructions (Chemometec).

### 4.6. Cytofluorometric Studies

Cytofluorometric acquisitions were performed by means of a FACSVerse cytofluorometer (BD Biosciences). Data was statistically evaluated using the Kaluza (Beckman Coulter) software. Only events characterized by normal forward scatter (FSC) and side scatter (SSC) parameters were gated for inclusion in the statistical analysis after exclusion of cell doublets.

#### 4.6.1. Cell Cycle Analysis

For the quantification of cell cycle profiling (DNA content), live cells were harvested, collected with the culture medium, and resuspended in 0.3 mL prewarmed growth medium supplemented with 2 mM Hoechst 33,342 (Sigma-Aldrich, Germany) for 30 min, at 37 °C, in a CO_2_ incubator. Cell suspensions were analyzed on a cytometer with ultraviolet excitation and emission collected at >450 nm. To quantify the apoptotic and DNA fragmented fraction, we measured the subG1 population of cells.

#### 4.6.2. Measurement of Cell Shrinkage

For the quantification of cell shrinkage, cells were harvested and collected with the culture medium before FACS assessment without any staining. Cell size was measured using the FSC and cell granularity using the SSC. Apoptotic cells are more granulated and smaller than live cells.

#### 4.6.3. Measurement of Cell Permeabilization

For the quantification of plasma membrane integrity, cells were harvested and collected with the culture medium and stained with 0.5 to 1 μg/mL 7-aminoactinomycin (7AAD, which only incorporates into dead cells, from ThermoFisher) for 30 min, at 37 °C, before FACS assessment.

#### 4.6.4. Measurement of Cell Scrambling and Phosphatidylserine Exposure

For the simultaneous quantification of plasma membrane integrity and phosphatidylserine exposure, cells were harvested and collected with the culture medium and stained with Annexin-V-FITC (1:200 dilution; ImmunoTools, Friesoythe, Germany) and 1 μg/mL PI (which only incorporates into dead cells, from Sigma-Aldrich) for 30 min, at 37 °C, before FACS assessment.

#### 4.6.5. Measurement of Mitochondria Accumulation

For staining mitochondria, cells were harvested and collected with the culture medium and labelled for 45 min, at 37 °C, with a 100 nM of MitoTracker Green MTG (ThermoFisher, Germany) before FACS assessment. The signal shift was measured comparatively to non-treated cells.

#### 4.6.6. Measurement of Intracellular Calcium Concentration

For the evaluation of cytosolic calcium, cells were collected and suspended in growth medium supplemented with 5 μM of the calcium tracker Fluo-3/AM (Biotium, Hayward, CA, USA). Cells were incubated at 37 °C, for 30 min, before calcium-dependent fluorescence intensity measurement. The Fluo-3/AM is measured with an excitation wavelength of 488 nm (blue laser) and an emission wavelength of 530 nm. The signal shift and the geometric mean were measured comparatively to non-treated cells.

### 4.7. RT2 Profiler PCR Array Analysis

Gene expression profiling related to human cancer drug resistance was performed using a 384-well RT2 Profiler PCR Array (PAHS-004Z, QIAGEN AB, Sollentuna, Sweden). The full list of human cancer drug resistance genes is presented in Appendix A. Following total RNA extraction, cDNA was synthesized from Be2c-parental, Be2c-VCR based on the manufacturer’s instructions. Eighty-four different genes involved in cancer cell drug resistance were analyzed based on SYBR Green real-time PCR using the QuantStudio™ 7 Flex (Applied Biosystems, CA, USA). Normalization was performed using the five different housekeeping genes included in the array and the fold change was calculated using the RT2 Profiler PCR Array Data Analysis from Qiagen (http://www.qiagen.com/geneglobe).

### 4.8. Gene Signature Validation

We used the significantly regulated genes from the Be2c-VCR (Group 1) versus SK-N-Be2c (Control) to perform k-means clustering (ten iteration rounds) to group samples of two public datasets (GSE16476 and GSE62564). These latter analyses were performed on the R2 web platform (http://r2.amc.nl). For each dataset, the overall and the event-free (relapse-free) survival probability were calculated by using the Kaplan–Meier method, and the significance of the difference between Kaplan–Meier curves was calculated by the log-rank test. The Cox regression model was used to test for independent predictive ability of chemoresistant gene signature after adjustment for the following other significant factors: MYCN amplification, age, and INSS stages. Statistical analyses were performed by SPSS software with statistical significance set at 5%.

### 4.9. Gene Ontology and Pathway Enrichment

Differentially expressed gene lists were used to perform gene ontology (GO) and pathway enrichment analysis by using the tool WebGestalt (www.webgestalt.org). We used the overrepresentation enrichment analysis method to query the GO and the KEGG databases with the default options. *p*-values were calculated with the hypergeometric test and statistical significance level was set at false discovery rate (FDR) ≤ 0.05.

### 4.10. Statistical Analyses

Statistical analyses were performed using the SigmaPlot or GraphPad Software. Results are expressed as mean ± SD, ±SEM, or as a percent. *p*-values * *p* < 0.05, ** *p* < 0.01, and *** *p* < 0.001 were deemed statistically significant. Statistical comparisons were assessed by analysis of variance (ANOVA) or by the Student’s *t*-test.

## Figures and Tables

**Figure 1 ijms-21-06811-f001:**
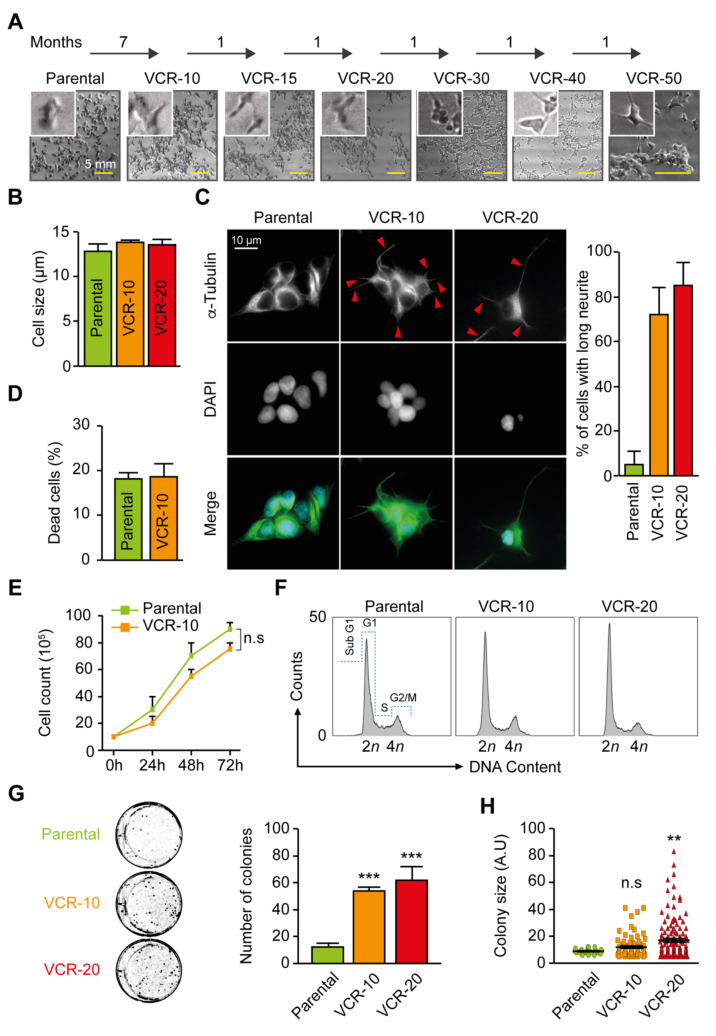
Establishment and characterization of vincristine-resistant neuroblastoma Be2c cells. (**A**) Neuroblastoma Be2c cells were cultured for 7 months with gradually increased concentrations of vincristine (VCR) (1–10 ng/mL). Cell lines with an even higher degree of resistance were subsequently created by additional subculturing in 15–50 ng/mL VCR containing media. Representative light microscopy images are reported. Scale bar = 5 mm; (**B**) Cell size was evaluated using the automatic cell counter Countess from Invitrogen. Quantitative results (mean ± SEM, *n* = 3) are shown; (**C**) Cell shape and cytoskeleton were evaluated by α-tubulin immunofluorescence staining. Neurites are indicated with red arrows. Neurite outgrowth is calculated by the number of cells with cell processes longer than the length of two cell bodies (mean ± SEM, *n* = 2); (**D**) Cell viability was assessed using propidium iodide staining (mean ± SEM, *n* = 3); (**E**) Cell proliferation was assessed using a cell counter (mean ± SEM, *n* = 3); (**F**) Representative cell cycle histograms For of parental, 10 and 20 ng/mL VCR-resistant cells. (**G**) Parental, 10 and 20 ng/mL VCR-resistant cells were seeded at a low cell density for clonogenic determinations. Representative wells from a 6-well plate with the different cell lines are shown. Graph displays averaged numbers of colonies from *n* = 3 (mean ± SEM); (**H**) Measurement of colony size comparing parental, 10 and 20 ng/mL VCR-resistant cells. (A.U. for arbitrary unit) (mean ± SEM, *n* = 3). n.s., non-significant. ** *p* < 0.01, *** *p* < 0.001 as compared with parental control cells (ANOVA, Tukey’s test).

**Figure 2 ijms-21-06811-f002:**
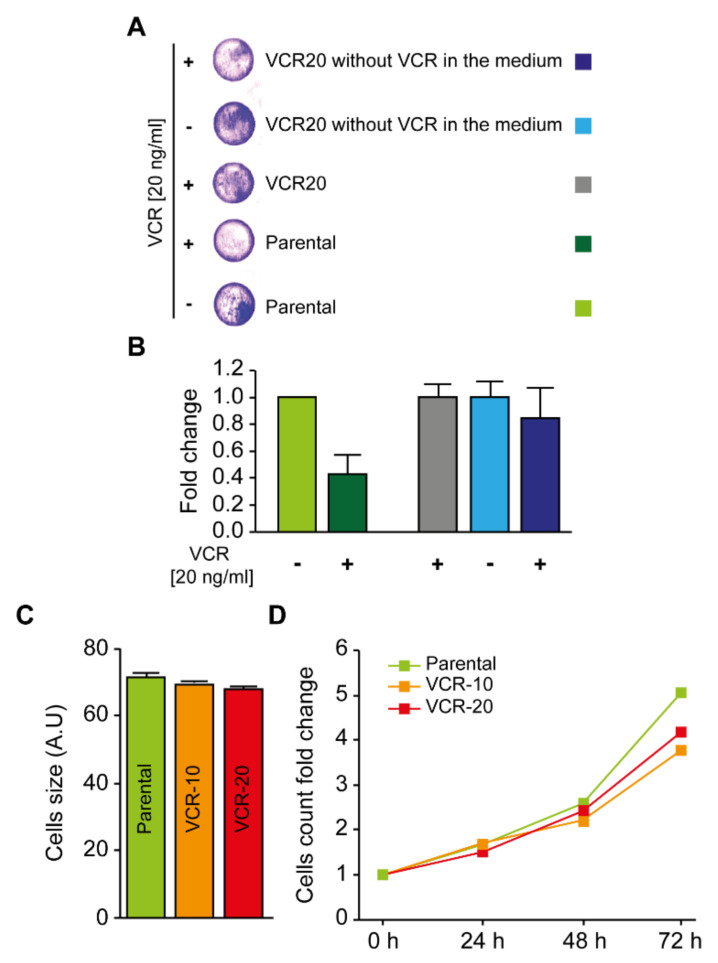
The mechanism of drug resistance is not immediately reversible upon cell release from drug. (**A**,**B**) Parental and 20 ng/mL VCR-resistant neuroblastoma cells were cultured for 3 weeks in a vincristine-free medium. Cells were seeded and treated for 48 h with 20 ng/mL vincristine for a crystal violet proliferation assay. Twenty ng/mL VCR-resistant cells kept in medium with vincristine were used as a control. Representative pictures of wells per condition are reported in (**A**), while the graph in (**B**) depicts the fold change in the absorbance at 595 nm of the treated cells as compared with non-treated cells. The green column illustrates the non-treated parental cells. The dark green column illustrates the parental cells treated with 20 ng/mL vincristine. The grey column illustrates the 20 ng/mL VCR-resistant cells kept in medium with vincristine. The blue column illustrates the 20 ng/mL VCR-resistant cells cultured for 3 weeks in vincristine-free medium. The dark blue column illustrates the 20 ng/mL VCR-resistant cells cultured for 3 weeks in vincristine-free medium, and then treated for 48 h with 20 ng/mL vincristine; (**C**) Parental, 10 and 20 ng/mL VCR-resistant neuroblastoma cells (labelled in green, orange, and red, respectively) were collected for cell size cytofluorometric assessment. Arithmetic means (±SEM, *n* = 3) of the cells forward scatter (FSC) are reported. Forward scatter is indicative of cell size; (**D**) Cells were seeded in 12-well plates at the exact density of 5 × 104 cells/well. Every 24 h, for 3 days, cells were collected and acquired by FACS. The corresponding number of events characterized by normal forward scatter (FSC) and side scatter (SSC) was determined.

**Figure 3 ijms-21-06811-f003:**
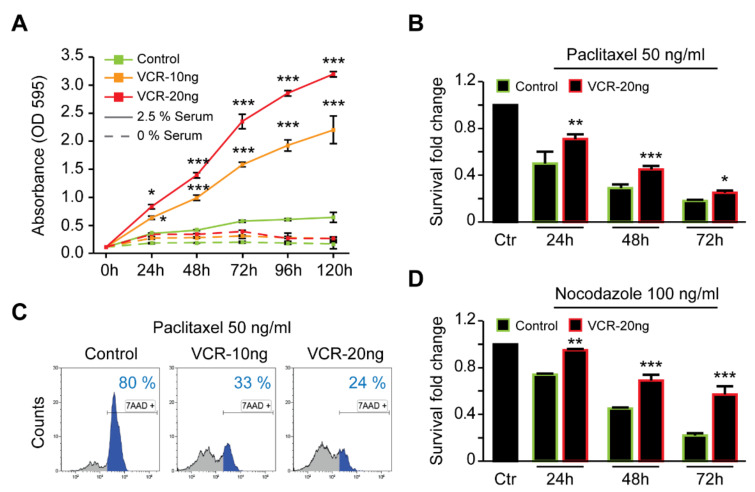
Vincristine-resistant neuroblastoma cells are more resistant to other microtubule-destabilizing drugs as compared with parental cells. (**A**) Parental, 10 and 20 ng/mL VCR-resistant neuroblastoma cells were cultured for 5 days with or without a low concentration of serum (2.5%). Cell proliferation was assessed using a crystal violet assay. Quantitative data are shown (mean ± SEM, *n* = 4). Full lines refer to the 2.5% serum condition while discontinued lines refer to cells grown in the complete absence of serum; (**B**) Parental and 20 ng/mL VCR-resistant cells (framed in green and red, respectively) were seeded at an initial density of 2000 cells per well in 96-well dishes, and then treated with 50 ng/mL paclitaxel. At the indicated time points (24, 48, and 72 h), cells were stained with crystal violet and the fraction of living cells was measured by absorbance at 595 nm. The bar framed in black represents the reference control treated with DMSO (mean ± SEM, *n* = 3); (**C**) Parental, 10 and 20 ng/mL VCR-resistant neuroblastoma cells were treated with 50 ng/mL paclitaxel collected after 24 h and labelled with the fluorescent intercalator 7-AAD for flow cytometry acquisition and cell death evaluation. Representative plots of treated cells are reported. The blue fractions represent 7-AAD positive (death) cells as compared with the DMSO-treated cells. Percentages of dead cells from one representative experiment are reported; (**D**) Parental and 20 ng/mL VCR-resistant cells (framed in green and red, respectively) were seeded at an initial density of 2000 cells per well in 96-well dishes, and then treated with 100 ng/mL nocodazole. At the indicated time points (24, 48, and 72 h, respectively), cells were stained with crystal violet and the fraction of surviving cells was measured by absorbance at 595 nm. The bar framed in black represents the reference control treated with DMSO (mean ± SEM, *n* = 3). *p*-values * *p* < 0.05, ** *p* < 0.01, and *** *p* < 0.001 were deemed statistically significant.

**Figure 4 ijms-21-06811-f004:**
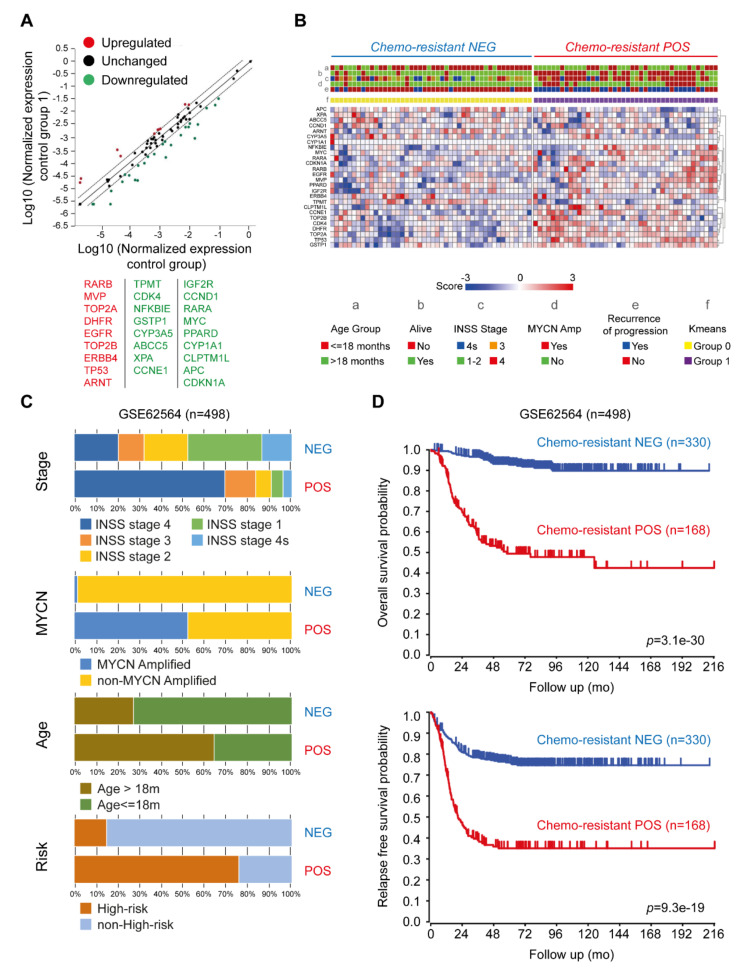
Expression profile of genes related to cancer drug resistance. (**A**) Scatter plot showing the expression profile of genes related to cancer drug resistance using the Human Cancer Drug Resistance RT^2^ Profiler PCR Array. The list of the genes is presented in Appendix A. The levels of relative expression for each gene in Be2c-VCR and Be2c-parental samples are plotted against each other in a log-log scatter plot after normalization was done using selected housekeeping genes. The line in the middle indicates relative fold changes. The stippled lines indicate the threshold for significant changes in gene expression, which was defined as twofold; (**B**) Gene expression heat map of the list of 26 genes found significantly regulated in the chemoresistance signature. The heat map is divided in two panels according to the results of ten rounds of k-means clustering steps. Clusters are depicted by the yellow-purple annotation track. We named ”chemoresistant NEG” (blue bar at the top) the Group 0 (yellow) and ”chemoresistant POS” (red bar at the top) the Group 1 (purple) of the k-means. Other annotation tracks reporting clinical information are explained in the color key at the bottom. Overall, the chemoresistant POS group shows higher expression of the chemoresistance signature upregulated genes and lower expression of the downregulated genes. In addition, the chemoresistant POS cluster shows patients with worse characteristics as compared with chemoresistant NEG; (**C**) The stacked bar plots show the phenotypic features of samples in GSE62564 stratified by k-means clusters. For this data sets we report the International Neuroblastoma Staging System (INSS) Stage, MYCN amplification status, age at diagnosis grouped by the 18 months cutoff, and the Risk status classification. (**D**) Kaplan–Meier curves calculated by the log-rank test shows the overall and the relapse-free survival probabilities for GSE62564. *p*-values were calculated with log-rank test. All the analyses were performed on the R2 web platform (http://r2.amc.nl).

**Figure 5 ijms-21-06811-f005:**
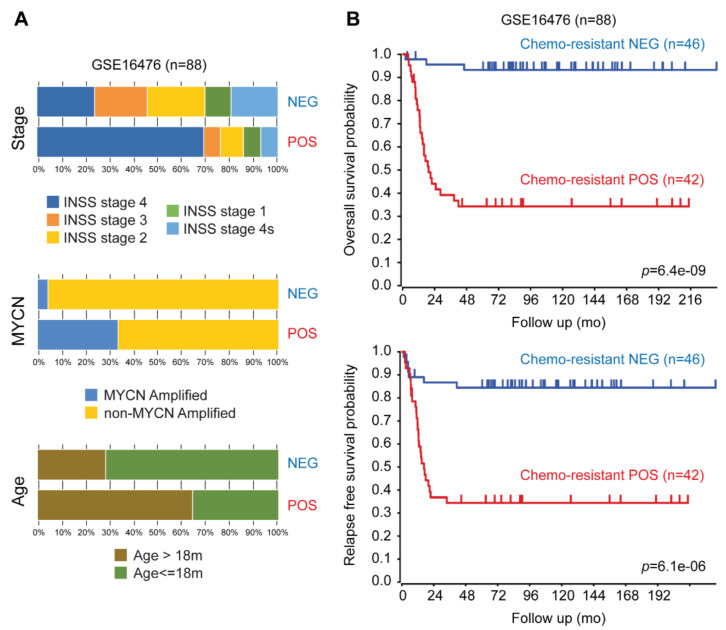
Chemoresistant POS groups have worse overall survival probabilities as compared with chemoresistant NEG and worse relapse-free survival probabilities over time. (**A**) The stacked bar plot shows the phenotypic features of samples in GSE16476 stratified by k-means clusters. For this dataset we report the International Neuroblastoma Staging System (INSS) Stage, MYCN amplification status, and age at diagnosis grouped by the 18 months cutoff. (**B**) Kaplan–Meier curves calculated by the log-rank test showing the overall and the relapse-free survival probabilities for GSE16476. *p*-values were calculated with log-rank test. All the analyses were performed on the R2 web platform (http://r2.amc.nl).

**Table 1 ijms-21-06811-t001:** Cell cycle distribution assessed by flow cytometry using parental, 10 or 20 ng/mL VCR-resistant cells and stained with Hoechst 33,342 (mean ± SEM, *n* = 3).

Cell Cycle Phases	Parental	VCR-10	VCR-20
SubG1	1.3 ± 0.6	1.0	0.7 ± 0.6
G1	57.7 ± 2.5	68.3 ± 2.5	67.3 ± 2.5
S	16.7 ± 1.2	13.0 ± 2.6	14.0 ± 1.7
G2/M	24.3 ± 3.2	17.3 ± 0.6	18.0 ± 1.0

**Table 2 ijms-21-06811-t002:** Half maximal inhibitory concentration (IC50 including SD values) of parental and 20 ng/mL VCR-resistant cells to a panel of commonly used drugs in chemotherapy. IC50 values were determined after 48 h of treatments using the alamarBlue cell viability assay.

Drugs	Parental	VCR 20 ng
Paclitaxel	32.11 ng/mL (± 1.33)	70.7 ng/mL (± 2.11)
Nocodazole	91.97 ng/mL (± 1.18)	337 ng/mL (± 7.64)
Cisplatin	1.37 µg/mL (± 0.27)	2.23 µg/mL (± 0.35)
Etoposide	1.6 µg/mL (± 0.26)	2.14 µg/mL (± 0.47)
Vincristine	0.47 ng/mL (± 0.01)	>700 ng/mL

**Table 3 ijms-21-06811-t003:** Multivariate Cox regression models for 498 NBs (GSE62564) and for 88 NBs (GSE16476) based on Overall Survival and the Relapse-Free Survival probabilities considering prognostic markers and the gene signature.

**GSE625664 (498 NBs)**	**Overall Survival**				**Relapse-Free Survival**			
**Markers**	***p*-Value**	**Hazard Ratio**	**95% CI**		***p*-Value**	**Hazard Ratio**	**95% CI**	
			**Lower**	**Upper**			**Lower**	**Upper**
INSS Stage (1, 2, 3, 4s vs. 4)	0.0004	2.62	1.54	4.45	0	2	1.39	2.9
MYCN (non-Ampl vs. Ampl)	0.0019	2.08	1.31	3.29	0.19	1.3	0.88	1.91
Age at diagnosis (<18m vs. ≥18m)	0.00001	3.38	1.96	5.81	0	1.95	1.38	2.77
Chemo-resistant (Neg vs. Pos)	0.0002	2.83	1.63	4.92	0	1.81	1.22	2.68
**GSE16476 (88 NBs)**	**Overall Survival**				**Event-Free Survival**			
**Markers**	***p*-Value**	**Hazard Ratio**	**95% CI**		***p*-Value**	**Hazard Ratio**	**95% CI**	
			**Lower**	**Upper**			**Lower**	**Upper**
INSS Stage (1, 2, 3, 4s vs. 4)	0.586	1.47	0.37	5.89	0.89	1.08	0.38	3.05
MYCN (non-Ampl vs. Ampl)	0.532	1.28	0.59	2.76	0.37	1.41	0.66	2.99
Age at diagnosis (<18m vs. ≥18m)	0.0009	39.08	4.5	339.56	0.001	6.09	2.09	17.74
Chemo-resistant (Neg vs. Pos)	0.0009	9.49	2.5	26.01	0.004	4.07	1.58	10.45

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
