# Peer review of "Gene Expression Signature of Acquired Chemoresistance in Neuroblastoma Cells"

_ijms, 2020, doi:10.3390/ijms21186811_

Round 1

Reviewer 1 Report

This manuscript examines the behaviour of neuroblastoma cell line SKNBE2(C) after they have been made increasingly resistant to the chemotherapeutic drug vincristine. The goal of the work was to define a gene expression signature in resistant cells that could potentially be used in a clinical setting to inform chemotherapy regimens.

The authors used a standard approach to increase drug resistance, using increasing levels of drug over a period of months in culture. They generated cells that were significantly more resistant to this drug and analysed these cells in terms of phenotype, proliferation and cell cycle patterns. They then examined the transcriptome of the parental and resistant derivatives and compared the expression differences to a set of known chemoresistance-related genes. A signature was thus defined and this was shown to correlate significantly with a range of patient and tumour parameters such as survival, MYCN amplification and tumour grades. The authors conclude that this signature could prove useful in predicting tumour behavior in response to chemotherapy independently of several known prognostic factors.  This gene signature is therefore potentially interesting and could be clinically relevant.

The study is well performed and the data are presented clearly with appropriate statistical analyses. Nevertheless, there are a number of points that need to be addressed with the research and with the manuscript content itself.

  1. Vincristine was used to develop the resistant cell lines but there could have been more information provided about SKNBe2C in terms of whether they are already partially resistant and how they compare to other cell lines. After all, SKNBe2C are from an aggressive, metastasising tumour.
  2. A good level of detail is given in describing the behaviour of the resistant cells. However, this appears incomplete. For example, Figure 1 shows neurite measurement data whereas the text ignores this. The text states the change in cell shape only. This must be re-written to reflect the data shown and what conclusion can be drawn in terms of cell differentiation.
  3. Figure 1A is not clear since the cells are too small. If the authors want to convince us there is a cell shape change, they need to show much enlarged phase contrast images.
  4. In Figure 1F, it would be better to tabulate the cell cycle stages instead of a stacked bar chart, as it is not possible to properly judge these data. Such a table should also have statistical analyses to demonstrate significance difference if they do/do not occur.
  5. The authors indicate that vincristine can give rise to resistance due to re-re-expression of fetal tubulins. However, they go on to show this is not the case here since the cells already express such a tubulin. Moreover they show that these cells are multiply resistant now to a range of drugs. This surely indicates that they are now multi-drug resistant and perhaps this is operating through a known mechanism such as drug efflux. This needs to be checked and discussed.
  6. The data from the other drugs tested is shown in Table 1, with IC50 values. First, some statistics would be useful in this table (eg SD values). Second, why is vincristine not shown here too as a comparator? This would be very useful as it would show the fold increase in resistant to this drug compared to the others.
  7. In the gene signature study the authors used a commercial screening platform. The authors should provide the exact product used and exactly what the 84 gene set entails (types of genes, cancers etc; is it Qiagen CATALOG No. - PAHS-004Z, PRODUCT No. – 330231?). This gene set could be given in a supplementary file.
  8. In many of the figures there is much use of coloured lines and bars. This provides impact, but it will also make it hard for those with colour blindness to understand the data. The colours should either be replaced with other more appropriate ones, or switched to grey scales.
  9. Is the Table 3 incomplete since there appear to items missing in the column “Markers”?
  10. In table 2, the percentage values for chemo-resistant NEG, GSE16476 do not add up to 1005 for the tumour stages. Also these data are the same as those shown in Figure 4C and 5A and it is unclear whether they add anything of value (they could be moved to a supplementary table).
  11. The authors indicate that their gene signature provides independent prognostic value. However, how specific is this to the use of vincristine? How does this signature relate to the original premise of using this drug as a starting point? As above, is this just a multi-drug resistance signature that might already exist? After all, the Qiagen gene set is already one such form of signature, albeit not neuroblastoma-focussed. The authors need to discuss this in more detail and must also provide more references to other related drug resistance gene/protein signature work in neuroblastoma (or lack thereof).
  12. The authors should also expand on their proposed usefulness of the gene signature. Would this be used before, during or after drug treatments, and would it be predictive, or instead more efficiently identify patients who were developing resistance?

Manuscript Problems.

  1. The methods section clearly does not relate correctly to this manuscript. There are several sections that are no relevant at all to this work and seem to be remnants of a previously submitted article. For example, in 4.1 a number of other cell lines are described in detail and these do not appear in the data. This whole section needs to be thoroughly edited.

Author Response

Response to Reviewer #1

Comments for the Authors

The study is well performed and the data are presented clearly with appropriate statistical analyses. Nevertheless, there are a number of points that need to be addressed with the research and with the manuscript content itself.

We thank the Reviewer for the careful review of our manuscript and the valuable and constructive suggestions.

  1. Vincristine was used to develop the resistant cell lines but there could have been more information provided about SKNBe2C in terms of whether they are already partially resistant and how they compare to other cell lines. After all, SKNBe2C are from an aggressive, metastasising

Response:

The reviewer is correct that SKNBe2C originates from aggressive and metastasising tumour. We have included additional information about this cell line (page 2). The reason for selecting this cell line was to have a cell line with MYCN amplification, p53 mutation and 1p deletion. SKNBe2C cell line compared to other NB cell lines were sensitive to 1 ng/ml Vincristine. Thereby, we needed to treat this cell line for 7 months with Vincristine to generate resistant cells.

  1. A good level of detail is given in describing the behaviour of the resistant cells. However, this appears incomplete. For example, Figure 1 shows neurite measurement data whereas the text ignores this. The text states the change in cell shape only. This must be re-written to reflect the data shown and what conclusion can be drawn in terms of cell

Response:

We have included description of elongated cells as neurite outgrowth (page 2) and explained that these changes can be considered as neuroblastoma cell differentiation and non-proliferative stage of the cells.

  1. Figure 1A is not clear since the cells are too small. If the authors want to convince us there is a cell shape change, they need to show much enlarged phase contrast Response:

We have inserted enlarged images as insert in Figure 1A.

  1. In Figure 1F, it would be better to tabulate the cell cycle stages instead of a stacked bar chart, as it is not possible to properly judge these Such a table should also have statistical analyses to demonstrate significance difference if they do/do not occur. Response:

We followed the reviewer suggestion and present the cell cycle stages as table with statistical analyses (Table 1).

  1. The authors indicate that vincristine can give rise to resistance due to re-re-expression of fetal tubulins. However, they go on to show this is not the case here since the cells already express such a tubulin. Moreover they show that these cells are multiply resistant now to a range of This surely indicates that they are now multi-drug resistant and perhaps this is operating through a known mechanism such as drug efflux. This needs to be checked and discussed.

Response:

The reviewer is correct. We expected increased expression of tubulin in resistant cells, but this was not altered in Be2c-VCR cells compared to parental cells, which as the reviewer suggest could be due to the other mechanism. Besides the tubulin blot in Fig. 3E, we are not addressing the mechanism for chemoresistance in our manuscript. Because of this, we decided to remove Figure 3E from the manuscript. Instead, we will investigate the mechanism of chemoresistance of these cells in more details in future studies. The following sentence was included in the discussion “However, further studies need to establish the precise mechanism behind chemoresistant neuroblastoma cells” (page 13).

  1. The data from the other drugs tested is shown in Table 1, with IC50 values. First, some statistics would be useful in this table (eg SD values). Second, why is vincristine not shown here too as a comparator? This would be very useful as it would show the fold increase in resistant to this drug compared to the

Response:

We have included SD values and vincristine treatment in the previous Table 1 (now presented as Table 2).

  1. In the gene signature study the authors used a commercial screening platform. The authors should provide the exact product used and exactly what the 84 gene set entails (types of genes, cancers etc; is it Qiagen CATALOG No. - PAHS-004Z, PRODUCT No. – 330231?). This gene set could be given in a supplementary

Response:

We have included this information as Supplementary Fig. 1 (page 16).

  1. In many of the figures there is much use of coloured lines and bars. This provides impact, but it will also make it hard for those with colour blindness to understand the data. The colours should either be replaced with other more appropriate ones or switched to grey

Response:

We have changed the color of the bars to black and thickened the lines for the bars in figure 3B and 3D. If the reviewer agrees, we suggest keeping the color code of the figures for the rest of the manuscript, because of the complexity of the analyzed materials including cell lines, concentrations of treatment, and time points.

  1. Is the Table 3 incomplete since there appear to items missing in the column “Markers”?

Response:

We thank the reviewer to have highlighted this error. The Table 3 was erroneously formatted after the conversion from word to pdf file. We have now solved this problem and adjusted the Table 3.

  1. In table 2, the percentage values for chemo-resistant NEG, GSE16476 do not add up to 1005 for the tumour Also these data are the same as those shown in Figure 4C and 5A and it is unclear whether they add anything of value (they could be moved to a supplementary table).

Response:

We agree with the Reviewer and removed Table 2 from the manuscript.

  1. The authors indicate that their gene signature provides independent prognostic value. However, how specific is this to the use of vincristine? How does this signature relate to the original premise of using this drug as a starting point? As above, is this just a multi-drug resistance signature that might already exist? After all, the Qiagen gene set is already one such form of signature, albeit not neuroblastoma-focussed. The authors need to discuss this in more detail and must also provide more references to other related drug resistance gene/protein signature work in neuroblastoma (or lack thereof).
  2. The authors should also expand on their proposed usefulness of the gene Would this be used before, during or after drug treatments, and would it be predictive, or instead more efficiently identify patients who were developing resistance? Response to question 11 and 12:

We agree with the Reviewer. We have now better discussed the significance of our gene signature and added comments on previously published gene-based classifier in NB (page 13).

Manuscript Problems.

  1. The methods section clearly does not relate correctly to this manuscript. There are several sections that are no relevant at all to this work and seem to be remnants of a previously submitted article. For example, in 4.1 a number of other cell lines are described in detail and these do not appear in the data. This whole section needs to be thoroughly edited.

Response:

We apologize for this mistake. Material and Method section have been thoroughly revised.

Reviewer 2 Report

Drug resistance of childhood cancer neuroblastoma is a major clinical problem.  In the present study, the Authros aimed at identifying chemoresistance gene expression signatures in vincristine resistant neuroblastoma cells. To do so they investigated the expression profiles of genes involved in cancer drug resistance. This specific gene  expression patterns could be useful to predict the behavior of a tumor in response to chemotherapy and for  predicting the prognosis of high-risk neuroblastoma patients. This signature could help tailored   chemotherapeutic approaches in neuroblastoma patients minimizing development of resistance

This is a very interesting paper. I have a few comments on it as it stands:

  1. One of the groups colaborating to this paper has previously shown the relevance of multidrag resistant proteins in development of chemoresistance to vincristine of NB cell lines in vitro and has shown that this may be efficiently counteracted by using a nitric oxide derivate of Saquinavir, named Saquinavir-NO that exerts strong chemotherapeutic properties. The Authors should quote and dicuss this finding in the context of this paper and indicate that tailored chemotherapeutic approaches screend for ability to overcome chemoresistance should be particularly consiedered in light of this paper.

Rothweiler F, Michaelis M, Brauer P, Otte J, Weber K, Fehse B, Doerr HW, Wiese M, Kreuter J, Al-Abed Y, Nicoletti F, Cinatl J Jr.Anticancer effects of the nitric oxide-modified saquinavir derivative saquinavir-NO against multidrug-resistant cancer cells.Neoplasia. 2010 Dec;12(12):1023-30. doi: 10.1593/neo.10856.

    Maksimovic-Ivanic D, Fagone P, McCubrey J, Bendtzen K, Mijatovic S, Nicoletti F. HIV-protease inhibitors for the treatment of cancer: Repositioning HIV protease inhibitors while developing more potent NO-hybridized derivatives?Int J Cancer. 2017 Apr 15;140(8):1713-1726. doi: 10.1002/ijc.30529. Epub 2017 Jan 20.   2. In addition to  other oncogenic pathways that may be responsible of developmet of chemoresistance the Authors should also quote and dicuss the emerging evidence indicating a key  role played in neuroblastoma from macrophage migration inhibitory factor and its homologue DDT     
Cavalli E, Ciurleo R, Petralia MC, Fagone P, Bella R, Mangano K, Nicoletti F, Bramanti P, Basile MS. Emerging Role of the Macrophage Migration Inhibitory Factor Family of Cytokines in Neuroblastoma. Pathogenic Effectors and Novel Therapeutic Targets?Molecules. 2020 Mar 6;25(5):1194. doi: 10.3390/molecules25051194.     Cavalli E, Mazzon E, Mammana S, Basile MS, Lombardo SD, Mangano K, Bramanti P, Nicoletti F, Fagone P, Petralia MC.Overexpression of Macrophage Migration Inhibitory Factor and Its Homologue D-Dopachrome Tautomerase as Negative Prognostic Factor in Neuroblastoma.Brain Sci. 2019 Oct 19;9(10):284. doi: 10.3390/brainsci9100284.   In search of completeness of information the Authors should also emphasize that MIF and DDT appear to be implicated not only in neuroblastoma development but also in other forms of cancer and in different pathologies including autoimmune diseases and pshychiatric disorders       Koh HM, Kim DC.Prognostic significance of macrophage migration inhibitory factor expression in cancer patients: A systematic review and meta-analysis.Medicine (Baltimore). 2020 Aug 7;99(32):e21575. doi: 10.1097/MD.0000000000021575.

Sven Günther Paolo Fagone Gaël Jalce Atanas G Atanasov Christophe Guignabert Ferdinando Nicoletti Role of MIF and D-DT in immune-inflammatory, autoimmune, and chronic respiratory diseases: from pathogenic factors to therapeutic targets Drugs Discovery Today 2019 Feb;24(2):428-439.  doi: 10.1016/j.drudis.2018.11.003. Epub 2018 Nov 13.
Petralia MC, Mazzon E, Fagone P, Basile MS, Lenzo V, Quattropani MC, Bendtzen K, Nicoletti F.Pathogenic contribution of the Macrophage migration inhibitory factor family to major depressive disorder and emerging tailored therapeutic approaches. J Affect Disord. 2020 Feb 15;263:15-24. doi: 10.1016/j.jad.2019.11.127. Epub 2019 Nov 30.

Author Response

Response to Reviewer #2

Comments for the Authors

Drug resistance of childhood cancer neuroblastoma is a major clinical problem. In the present study, the Authros aimed at identifying chemoresistance gene expression signatures in vincristine resistant neuroblastoma cells. To do so they investigated the expression profiles of genes involved in cancer drug resistance. This specific gene expression patterns could be useful to predict the behavior of a tumor in response to chemotherapy and for predicting the prognosis of high-risk neuroblastoma patients. This signature could help tailored chemotherapeutic approaches in neuroblastoma patients minimizing development of resistance

This is a very interesting paper. I have a few comments on it as it stands: Response:

We thank the Reviewer for the careful review of our manuscript and the valuable and constructive suggestions.

One of the groups colaborating to this paper has previously shown the relevance of multidrag resistant proteins in development of chemoresistance to vincristine of NB cell lines in vitro and has shown that this may be efficiently counteracted by using a nitric oxide derivate of Saquinavir, named Saquinavir-NO that exerts strong chemotherapeutic properties. The Authors should quote and dicuss this finding in the context of this paper and indicate that tailored chemotherapeutic approaches screend for ability to overcome chemoresistance should be particularly consiedered in light of this paper.

Rothweiler F, Michaelis M, Brauer P, Otte J, Weber K, Fehse B, Doerr HW, Wiese M, Kreuter J, Al-Abed Y, Nicoletti F, Cinatl J Jr. Anticancer effects of the nitric oxide- modified saquinavir derivative saquinavir-NO against multidrug-resistant cancer cells. Neoplasia. 2010 Dec;12(12):1023-30. doi: 10.1593/neo.10856.

Maksimovic-Ivanic D, Fagone P, McCubrey J, Bendtzen K, Mijatovic S, Nicoletti F. HIV-protease inhibitors for the treatment of cancer: Repositioning HIV protease inhibitors while developing more potent NO-hybridized derivatives?Int J Cancer. 2017 Apr 15;140(8):1713-1726. doi: 10.1002/ijc.30529. Epub 2017 Jan 20.

Response:

Since our manuscript is not dealing with the discovery of drug against chemoresistance cancer cells, these two publications will not fit in the scope of the paper. We are now in the progress of revision of another manuscript from our lab. entitled “Discovery of epi-enprioline as a novel drug for the treatment of vincristine resistant neuroblastoma”, where we believe that these two references could fit very well.

In addition to other oncogenic pathways that may be responsible of developmet of chemoresistance the Authors should also quote and dicuss the emerging evidence indicating a key role played in neuroblastoma from macrophage migration inhibitory factor and its homologue DDT.

Cavalli E, Ciurleo R, Petralia MC, Fagone P, Bella R, Mangano K, Nicoletti F, Bramanti P, Basile MS. Emerging Role of the Macrophage Migration Inhibitory Factor Family of Cytokines in Neuroblastoma. Pathogenic Effectors and Novel Therapeutic Targets?Molecules. 2020 Mar 6;25(5):1194. doi: 10.3390/molecules25051194.

Cavalli E, Mazzon E, Mammana S, Basile MS, Lombardo SD, Mangano K, Bramanti P, Nicoletti F, Fagone P, Petralia MC. Overexpression of Macrophage Migration Inhibitory Factor and Its Homologue D-Dopachrome Tautomerase as Negative Prognostic Factor in Neuroblastoma. Brain Sci. 2019 Oct 19;9(10):284. doi: 10.3390/brainsci9100284.

In search of completeness of information the Authors should also emphasize that MIF and DDT appear to be implicated not only in neuroblastoma development but also in other forms of cancer and in different pathologies including autoimmune diseases and pshychiatric disorders

Koh HM, Kim DC. Prognostic significance of macrophage migration inhibitory factor expression in cancer patients: A systematic review and meta-analysis. Medicine (Baltimore). 2020 Aug 7;99(32):e21575. doi: 10.1097/MD.0000000000021575. Sven

Günther , Paolo Fagone , Gaël Jalce , Atanas G Atanasov , Christophe Guignabert , Ferdinando Nicoletti Role of MIF and D-DT in immune-inflammatory, autoimmune, and chronic respiratory diseases: from pathogenic factors to therapeutic targets Drugs Discovery Today 2019 Feb;24(2):428-439. doi: 10.1016/j.drudis.2018.11.003. Epub 2018 Nov 13.

Petralia MC, Mazzon E, Fagone P, Basile MS, Lenzo V, Quattropani MC, Bendtzen K, Nicoletti F. Pathogenic contribution of the Macrophage migration inhibitory factor

family to major depressive disorder and emerging tailored therapeutic approaches. J Affect Disord. 2020 Feb 15;263:15-24. doi: 10.1016/j.jad.2019.11.127. Epub 2019 Nov 30.

Response:

We would like to thank the reviewer for suggesting these refences and we tried to include the suggested references in our manuscript without any success. Unfortunately, these studies were not fitting within the scope of the present manuscript; chemoresistance in neuroblastoma cells. However, these references will be used in our future studies concerning development of novel drugs and treatments for NB.

We are now in the progress of revision of another manuscript from our lab. entitled “Discovery of epi-enprioline as a novel drug for the treatment of vincristine resistant neuroblastoma”, where we believe that these references could fit very well.
